# GAGAM v1.2: An Improvement on Peak Labeling and Genomic Annotated Gene Activity Matrix Construction

**DOI:** 10.3390/genes14010115

**Published:** 2022-12-30

**Authors:** Lorenzo Martini, Roberta Bardini, Alessandro Savino, Stefano Di Carlo

**Affiliations:** Politecnico di Torino, Control and Computer Engineering Department, 10129 Torino, Italy

**Keywords:** epigenomic single-cell data, gene activity matrix, bioinformatics

## Abstract

Single-cell Assay for Transposase-Accessible Chromatin using sequencing (scATAC-seq) is rapidly becoming a powerful technology for assessing the epigenetic landscape of thousands of cells. However, the sparsity of the resulting data poses significant challenges to their interpretability and informativeness. Different computational methods are available, proposing ways to generate significant features from accessibility data and process them to obtain meaningful results. Foremost among them is the peak calling, which interprets the raw scATAC-seq data generating the peaks as features. However, scATAC-seq data are not trivially comparable with single-cell RNA sequencing (scRNA-seq) data, an increasingly pressing challenge since the necessity of multimodal experiments integration. For this reason, this study wants to improve the concept of the Gene Activity Matrix (GAM), which links the accessibility data to the genes, by proposing an improved version of the Genomic-Annotated Gene Activity Matrix (GAGAM) concept. Specifically, this paper presents GAGAM v1.2, a new and better version of GAGAM v1.0. GAGAM aims to label the peaks and link them to the genes through functional annotation of the whole genome. Using genes as features in scATAC-seq datasets makes different datasets comparable and allows linking gene accessibility and expression. This link is crucial for gene regulation understanding and fundamental for the increasing impact of multi-omics data. Results confirm that our method performs better than the previous GAMs and shows a preliminary comparison with scRNA-seq data.

## 1. Introduction

Recent advances in Next Generation Sequencing (NGS) technologies paved the way for single-cell multi-omics data analysis, which captures different facets of cells’ regulative state, including the epigenome, the genome, the transcriptome, and the proteome [1]. Multi-omics approaches increase resolution and sensitivity in the characterization of cellular states, the identification of known or new cellular phenotypes, and the understanding of cell dynamics [2], supporting a quantitative and a comprehensive approach to studying cellular heterogeneity [3].

In particular, the combination of transcriptomic and epigenomic data provides integrated information on the functional activation of genes and the structural organization of chromatin. There are different experimental approaches to generating epigenomic data. These include accessibility measurements, which indicate whether chromatin is open or closed at genomic locations, exposing other genomic regions for transcriptional and regulatory processes [4]. These data have a very different organization than transcriptomic data indicating the expression level of genes.

In general, analyzing data simultaneously from multiple omics does not guarantee a gain in the informativeness of the analysis of the cellular system. Different mechanisms underlie each omic layer, and complex relations link different layers. Purely data-driven multi-omic combinations overlook the connections between omic layers. These methods fail to provide a systemic analysis and can also lead to inaccurate assumptions on such relations [5]. Furthermore, it is well-known that expression levels show almost no correlation to protein quantity in the cell. This is due to complex and non-obvious regulatory dynamics between transcriptional and translational regulations. In these dynamics, it is not sure that a large pool of mRNA for a gene will yield a large quantity of protein and vice-versa. Due to the uncertainty and complexity of biological dynamics, it is not trivial to treat transcriptomic and proteomic data under the assumption that the way they measure functional activation is comparable. Epigenomic data allow us to extrapolate a measure of transcriptional activity at the relevant regions for specific genes. This is often considered a predictor of the expression levels of the same gene. Yet, drawing a linear relation between activity and expression levels is far from trivial. Accessibility is a necessary but not sufficient condition for a gene for its transcription: a completely accessible gene can show a full range of expression levels, including null expression. This does not support the assumption that the relation between accessibility and expression is linear in multi-omics analyses [6]. Thus, building better models of the relations among the different omic layers is pivotal to truly leverage multi-omics analysis and gain a systemic understanding of the cellular state. Multi-omics analysis strategies must combine data-driven and model-driven approaches to account for the complex relations among different omic layers explicitly and formally [5].

Focusing on the relations between the epigenomic and transcriptomic layers, the concept of gene activity, i.e., the overall accessibility of a gene allowing its transcription inside the cell [7], is an example of a data- and model-driven approach which facilitates the joint analysis of epigenomic and transcriptomic data. Chromatin accessibility refers to the structural organization of the DNA strands in eukaryotic genomes. Chromatin organization changes across cell types and cell differentiation phases. On a genome-wide scale, epigenetic regulations, by affecting chromatin openness, determine the accessibility of genomic regions to transcriptional processes. Complex patterns of chromatin occupancy by such epigenetic factors determine inverse patterns of chromatin accessibility, yielding structural information about binding of the transcriptional machinery and transcriptional regulators at the DNA [8]. Accessible regions can occur on coding regions (i.e., exons or introns), promoters, Transcription Starting Sites (TSS), and cis- and trans-regulatory elements, such as enhancers. Each of these types of regions has a different implication on the activity of a gene, linked to the functional effect determined by the effective binding of their specific binding proteins. After the assumption that regulatory elements that exhibit higher effective activation also show higher accessibility, epigenomic data can support the genome-wide identification of genomic regions that can discriminate different cell types or states at a functional level [9]. While epigenetic regulation can completely subtract genomic portions from transcriptional activity, an open genomic region undergoes more nuanced epigenetic regulations with both positive and negative implications for transcriptional activity. Accessibility patterns can make the same region more or less accessible and, depending on the functionality of accessible elements, have different functional consequences on the transcriptional process. Considering the fundamental limitation that accessibility is a necessary but not sufficient condition for a gene to be transcribed and acknowledging the different contributions of coding and non-coding accessible regions to the activity of a gene provide suitable premises to build a model of gene activity to support the multimodal analysis of epigenomic and transcriptomic data.

Current approaches build a bridge between transcriptomics and epigenomics, supporting a direct comparison of gene activity and gene expression levels. In particular, GAMs [7] provide an effective way to summarize accessibility information deriving from single-cell experiments in a score for gene activity. A GAM, similarly to an expression matrix, has a subset of genes from the reference genome as features and cells as data points. Each element of the matrix ($GAM_{g,c}$) represents the Gene Activity Score (GAS) of a gene *g* in a cell *c* [7], based on the gene activity model of choice. Using the same set of genes in expression and activity matrices makes transcriptomic and epigenomic data easy to analyze with the same pipelines. Yet, as better explained in Section 2, even if current approaches to compute gene activity in GAMs aim to derive a measure of activity from accessibility data, they show limitations in capturing the contextual meaning and the regulative implications of epigenomic data. This hampers the consistent integration of activity and expression and, thus, a robust approach to epigenomic and transcriptomic multimodal analysis.

To overcome this limitation, this work introduces Genomic-Annotated Gene Activity Matrix (GAGAM) v1.2, a data- and model-driven analysis strategy for epigenomic data. It combines accessibility data with genomic annotations. This combination allows us to weigh the contributions of different accessible regions to the overall gene activity and functional significance. This model is an extension of a preliminary model published by Martini et al. (GAGAM v1.0) that was already an improvement to other GAM-based approaches. GAGAM v1.0 included an explicit, more comprehensive, and modular model of the different contributions to gene activity from the different annotated region types [10]. GAGAM v1.2 improves differential activity analyses’ resolution, explainability, and interpretability, supporting the study of cellular heterogeneity based on accessibility data alone. The GAS model underlying GAGAM v1.2 improves the precision in weighting contributions of accessible regions and the specificity of their attribution to each gene in the matrix. The capability of GAGAM v1.2 was extensively validated, showing improved performance compared to GAGAM v1.0 on the same datasets and comparable performance on newly acquired datasets. Moreover, this work provides preliminary results on the analytical comparison between epigenomic and transcriptomic data based on GAGAM v1.2 and expression matrices from multimodal datasets.

## 2. Background

The availability of data from multiple omic pools is the enabler of multimodal single-cell analysis and catalyzed a revolution in medicine and biology. These data paved the way for integrated, systemic, and data-driven approaches to biological complexity [5,11]. The multimodal analysis includes the integration of data from different experiments considering different omics, as well as experiments using multi-omic sequencing technologies. Multi-omic sequencing technologies sample multiple omic pools from the same cells, characterizing their state more comprehensively than the sequencing of single omics. Integrating information from more than one omic pool in the same cell ensures higher consistency, increasing reliability and accuracy of cellular heterogeneity studies, cell type identification, and inference of differentiation trajectories.

While gaining information richness, multi-omic approaches can lose throughput. As a result, their overall throughput is bound to the lowest throughput among the sequencing technologies employed.

Considering the integration of transcriptomic and epigenomic data, this issue emerges clearly. Single-cell transcriptomics is the most mature among single-cell methods. As illustrated in [12], sequencing technologies for single-cell transcriptomics relate to methods from the 1990s, such as single-cell quantitative polymerase chain reaction (qPCR) [13], based on the consideration of a small number of genes. However, the first example of single-cell transcriptomics considering large sets of genes dates back to 2009 [14]. During the last decade, technological advances in microfluidics and automation allowed extending the sequencing process to thousands of cells per experiment. While scRNA-seq can scale to hundreds of thousands of cells, most single-cell epigenomic sequencing technologies tend to have lower throughput [4]. In general, available single-cell epigenomics considers either the methylation pattern at the DNA or the chromatin state [15]. Single-cell bisulfite sequencing (scBS-seq) addresses methylation profiles in single cells [16]. Methylation of CpG sites often indicates repression of gene transcription, but methylation of the gene body links to transcriptional activation. Single-cell methylation profiling requires high numbers of cells in input, yet low throughput (tens to thousands of cells) [17]. Chromatin immunoprecipitation followed by sequencing (ChIP-seq) measures the actual presence of transcriptional regulators at the genome, capturing not only genomic accessibility but the presence of the actors of the transcriptional process as well [18]. Yet, data generated by single-cell ChIP-seq of individual cells are very sparse: downstream analyses require an experiment to interrogate thousands of cells to ensure robust results [19]. To reduce the number of input cells required, alternative methods to ChIP-seq, not depending on immunoprecipitation, but on a reaction occurring within the nucleus, such as CUT&Tag [20], have been developed [21]. scATAC-seq [22] measures chromatin accessibility profiles. As previously discussed, accessibility indicates an open state of a genomic region, which often implies that the region - either coding or non-coding - is available for the transcriptional machinery or its regulators to bind and exert their function. scATAC-seq is rapidly becoming the primary way to assess the accessibility of the whole genome at the single-cell resolution, supporting the interrogation of numbers of cells ranging from hundreds (plate-based technologies) [23] to tens of thousands (droplet-based technologies) [24]. Analysis techniques for scATAC-seq datasets employ different ways to define meaningful features for analyzing accessibility profiles, as shown in [25]. One of the most popular is the “peak calling,” which defines peaks, i.e., intervals on the genome that have a local enrichment of transposase cut-sites, meaning they are highly accessible [26]. This organizes accessibility data referring them to specific genomic regions. Yet, peaks come from an experiment-dependent set of chromosomal regions, which impedes direct comparison between epigenomic and transcriptomic datasets.

As described before, a GAM effectively defines robust accessibility features. The GAM considers the overall accessibility of the genomic regions linked to specific genes as defined in single-cell transcriptomics. Using scATAC-seq data, the GASs composing the elements of a GAM can be computed after the accessibility of the peaks related to the corresponding genomic region to such genes. However, the way to link peaks to the correct genetic region on the genome is not unique. The literature reports three main GAM-based strategies:GeneScoring sums the peaks in a broad region before and after a gene’s TSS, weighted by their distance from it [27]. This is the easiest way to define the activity of a gene. Nevertheless, it is simplistic since it overlooks the precise location of the genomic region relative to the gene and the distinction between coding and non-coding regulatory elements.Cicero defines the activity of a gene as the accessibility of the peaks overlapping the TSS and the accessibility of all the co-accessible peaks, i.e., the peaks whose accessibility has a high correlation among cells. This definition supports the claim that these peaks participate in transcriptional regulation of the genomic region [7]. This method, compared with GeneScoring, implements a strategy to factor in regulatory elements at the DNA as well. However, the way it identifies the genes is very simplistic since it relies on a single DNA base, i.e., the TSS. This limits the approach’s effectiveness in clearly linking genomic regions to genes. Moreover, co-accessibility evaluation is a very long and computationally heavy process. The GAS estimation does not consider the meaningfulness of the peaks, meaning if the co-accessible peaks are regulatory regions.Signac builds a GAM by counting all the raw fragment reads (coming from the experimental Tn5 Transposase insertions) overlapping the gene body [28]. The main limitation of this method is that it overlooks the functional significance of specific reads completely. It requires a fragment file related to the dataset containing all the fragments in each cell. It is a large file and rarely available, thus making the computation often impossible.

In general, all these methods oversimplify the relationship between a gene and the accessibility of its genomic region. The accessibility profile of a genomic region provides information about the regulation and resulting expression of the corresponding gene. However, this does not imply a direct nor linear relation between region accessibility and gene expression levels. If a gene is accessible, it is not necessarily expressed: accessibility only gives an insight into chromatin structure, determining whether transcription is possible.

The rise of multi-omic approaches - comparing single-omic information and measuring multiple omic pools in the same experiment - makes this point even more relevant. In particular, the emergence of new multi-omic NGS techniques allows performing both scATAC-seq and scRNA-seq simultaneously, posing the challenge of effectively analyzing and comparing information from transcriptomics and epigenomics. In this case, one way to achieve multi-omic consistent integration is to employ a model-driven approach for GAS computation.

To work towards this goal, we previously introduced in [10] the concept of Genomic-Annotated Gene Activity Matrix (GAGAM): a new, model-driven way to construct a GAM based on the functional genomic annotation of the peaks. A GAGAM differs from purely data-driven state-of-the-art scATAC-seq annotation methodologies since it considers that the contributions of different sets of accessible regulatory elements have a different meaning and impact on the GAM computation. This provides a methodological innovation. Beyond the bare superposition of accessibility data with genomic annotations, a GAGAM captures the functional impact of selected sets of accessible regions. This model builds an additional information layer on the genomic accessible regions not provided by routine pipelines for scATAC-seq annotation. The choices made to build the genomic model at the base of the GAGAM reflect the objective of multimodal epigenomic and transcriptomic single-cell data analysis. A pivotal point to support multimodal analysis is to allow comparison and joint analysis of different omic pools provided their structural differences.

This work introduces GAGAM v1.2, an improved version of GAGAM v1.0, to enhance the integration of accessibility and transcriptomic data. In GAGAM v1.2, the underlying gene activity model relies on an enhanced peak labeling process that focuses on the protein-coding gene regions that are more important from a transcriptomic standpoint. GAGAM v1.2 considers the contribution of the peaks inside the gene body more strictly than GAGAM v.1.0. Instead of accounting for the whole gene body (meaning both exon and intron parts), GAGAM v1.2 focuses on exon regions since they are the coding regions that are important from a transcriptomic standpoint. Moreover, during annotation, the identification of promoter peaks has been improved in GAGAM v1.2 using two approaches: the first based on the promoter signature of the candidate cis-Regulatory Elements (cCREs) tracks, the second on peaks overlapping the genes’ TSS. Eventually, GAGAM v1.2 introduces a normalization of the different contributions composing the final matrix scores not available in GAGAM v1.0 and improves the quality of the computed scores.

Compared to GAGAM v1.0, this work advances the GAGAM pipeline in methodology and performance. In particular, it implements specific strategies to capture cellular heterogeneity better, yielding improved performances. In addition, it implements an innovative version of the genomic model to allow direct qualitative comparison between GAGAM-based activity and gene expression for multimodal datasets.

## 3. Materials and Methods

Figure 1 introduces the workflow for computation and evaluation of GAGAM starting from a scATAC-seq dataset. Throughout the paper, the acronym GAGAM refers to the generic concept of the Genomic-Annotated Gene Activity Matrix. A specific version identifier allows us to refer to features implemented in one variant of GAGAM.

A scATAC-seq dataset contains a set *P* of peaks observed in a group of *C* cells. Each peak corresponds to a region of the target genome and is defined by its chromosome and a genomic coordinate pair p=(ch,start,stop). The dataset is a binary matrix $D_{\left|P|\times |C\right|}$ where rows are associated with peaks and columns with cells. An element of D equal to 1 denotes a peak (row) accessible in a cell (column).

The main contribution of GAGAM is to exploit information regarding overlaps of peaks, gene bodies, and genetic regulatory regions (i.e., promoters and enhancers) to build a GAM with higher information content.

### 3.1. Genomic Annotation

Understanding the biological meaning of the genomic region underlined by the peaks is fundamental for their proper model-driven usage. Therefore, the genomic annotation of peaks is the first step to constructing GAGAM. This process aims to enrich information regarding peaks with data from different genomic annotations useful for a model-driven construction of a GAM. Figure 2 shows the genomic model considered in this work. It represents a genomic unit, which includes three parts: (i) the gene body region, divided into Exons (the coding parts) and Introns (the non-coding parts), starting at the TSS, (ii) the gene Promoter, preceding the coding region, and (iii) a set of Enhancers that are distal to the gene.

The gene coding region is defined using NCBI RefSeq Genes [32] annotations, consisting of genes’ genomic coordinates. The NCBI RefSeq annotations are accessible using the NCBI Eukaryotic Genome Annotation Pipeline [33]. It consists of an annotated and curated information list of protein-coding and non-protein-coding genes.

Since GAGAM aims to link the epigenomic information to the transcriptomic one, it only considers the protein-coding gene regions. In GAGAM v1.2, pseudogenes, miRNA, and lncRNA regions, also present in the annotation, are discarded. This differs from GAGAM v1.0, which includes the lncRNA genes. LncRNA genes are coding regions but not for mRNA products measured in transcriptomic experiments and are not helpful for comparison with transcriptomic experiments. Every gene *g* in the target genome *G* is a tuple defining the gene’s chromosome and its genomic coordinates pair (i.e., g=(ch,start,stop)). From the same coordinates, one defines the TSS of a gene as a small region corresponding to the two bases of the first exon, i.e., $g_{TSS}=\left(ch,start,start+1\right)$.

Instead of accounting for the whole gene body (meaning both exon and intron parts) as in GAGAM v.1.0 [10], GAGAM v1.2 focuses on exon regions since they are the coding regions that are important from a transcriptomic standpoint. So for each gene *g*, there is a list of *n* exon regions defined by their coordinates, i.e., $g_{e_{1}}=\left(ch,start,stop_{e_{1}}\right)...g_{e_{n}}=\left(ch,start_{e_{n}},stop\right)$.

The regulative genomic regions are elements on the DNA footprints for the trans-acting proteins involved in transcription, either for the positioning of the basic transcriptional machinery or for the regulation. The annotation tracks are associations between a genomic region and a label indicating the function of the region. Given a target genome *G*, it is possible to define a set *R* of regulative regions with each region defined by the corresponding chromosome, the genomic coordinates pair, and a label (e.g., promoter or enhancer) indicating the function of the region (r=(ch,start,stop,l)).

Information regarding regulative gene regions is available from the Encyclopedia of DNA Elements (ENCODE) project, which provides an extensive collection of cell- and tissue-based genomic annotations tracks, including, for example, transcription, chromatin organization, epigenetic landscape dynamics, and protein binding sites from the mouse and human genomes [34]. ENCODE data are available through the ENCODE data portal [35].

This work only considers genomic annotations relative to promoter and enhancer functions. These regulatory elements are derived from the ENCODE cCREs. cCREs provides an extensive collection of annotated regions for the human and mouse genomes. Predicted classification of cCREs relies on biochemical signatures deriving from considering DNase hypersensitivity, histone methylation, acetylation, and CTCF binding data [34]. It is important to highlight that, given the current nature of the ENCODE annotation, it is impossible to get information for more complex regulatory elements. It is the case, for example, of the alternative promoters, which in the development and differentiation stages of some cell types allow for specific expression of alternate transcripts. Currently, GAGAM v1.2 does not handle alternative promoter usage since it considers a single promoter for each gene, thus excluding the contribution of alternative promoters either upstream or downstream of the TSS. To acknowledge different isoforms for each gene, it will be necessary to consider them as additional features, extending the structure of the GAGAM accordingly, which is not part of the current version.

In addition, since this work aims to label peaks from both human and mouse datasets, cCREs tracks (in BigBed format [36]) were collected from ENCODE for both the human [37] and mouse [38] genomes. To be precise, the cCREs tracks are only aligned to the most recent genome assemblies (hg38 for humans and mm10 for mice). Therefore, when dealing with data aligned to older assemblies, there is a need to convert peak coordinates to the track assembly. To achieve this, this work leverages the UCSC LiftOver tool [39] to map genomic coordinates of the peaks before performing peak labeling. This step can insert some uncertainty concerning the original data like some peaks cannot be aligned. However, this error is negligible since it regards just some tens of peaks, a small portion compared to the hundreds of thousands in the data.

The goal of the genomic annotation process is to associate each peak p∈P obtained from a scATAC-seq dataset D to a set of genomic annotation labels by analyzing how the peak overlaps with the different genomic regions.

GAGAM labels each peak p∈P with four possible tags: (1) prom for peaks overlapping a promoter region or a TSS, (2) enhD for peaks overlapping a distal enhancer region and not a promoter region, (3) exon for the peaks contained into a gene’s exon region, and (4) empty in all other cases. The rule to assign the label is summarized in the following equation:(1)$$\mathrm{PL}:p\in P\mapsto \;\left\{\begin{array}{cc} \mathrm{prom} & \mathrm{if}\;(\exists r\in R|r\subseteq p\wedge r_{l}=\mathrm{prom})\vee \\ \; & \left(\exists g_{tss}\in G|p\subseteq g_{tss}\right)\\ \mathrm{enhD} & \mathrm{if}\;(\exists r\in R|r\subseteq p\wedge r_{l}=\mathrm{enh})\wedge \\ \; & \left(∄r\in R|r\subseteq p\wedge r_{l}=\mathrm{prom}\right)\\ \mathrm{exon} & \mathrm{if}\;\exists g_{e}\in G|p\subseteq g_{e}\\ \mathrm{empty} & \mathrm{otherwise} \end{array}\right.$$

The operator a⊆b is used here to denote that the two regions *a* and *b* belong to the same chromosome with *b* overlapping *a* (i.e., $a_{start}\geq b_{start}\wedge a_{end}\leq b_{end}$). The computation of the intersection between peaks and annotation regions leverages the bigBedToBed tool from ENCODE [40].

Given the list of annotated peaks, GAGAM builds a gene activity matrix as a weighted sum of three separated matrices: (i) the promoter peaks matrix (P) indicating accessibility of genes associated with promoter peaks, (ii) the co-accessibility matrix (C) indicating the accessibility of genes associated with distal enhancer peaks, and (iii) the exon peaks matrix (E) indicating the accessibility of genes containing exon peaks.

The weights $w_{p}$, $w_{c}$, and $w_{e}$, as done in [10], are binary numbers that provide a formal and modular way to select which contributions to include in the final matrix computation. Therefore, these weights are not estimated from any training step but instead set at the beginning of the analysis. This work defines two combinations of weight values for GAGAM construction, one with all three contributions (i.e., $w_{p}=1$, $w_{c}=1$, and $w_{e}=1$), and one that excludes the contribution of exon peaks (i.e., $w_{p}=1$, $w_{c}=1$, and $w_{e}=0$)). In general, this allows us to study the contribution of each sub-matrix composing GAGAM to the gene activity and, consequentially, if some influence the gene expression in a specific way. This is not part of this specific paper but will be part of future analyses.

This yields a final curated and model-driven evaluation of the activity of the genes:(2)$$\mathrm{GAGAM}=w_{p}\cdot P+w_{c}\cdot C+w_{e}\cdot E$$

It is relevant to notice that, in GAGAM v1.0 [10], some peaks simultaneously overlapping enhancers signatures and intronic regions were contributing only to the Intragenic Matrix (the old version of the Exon Matrix). Thus their regulative information was lost. GAGAM v1.2 labels these peaks more precisely as enhD peaks, contributing to the co-accessibility Matrix.

The following paragraphs illustrate the modeling assumptions and construction process for each contribution to GAGAM.

### 3.2. Promoter Peaks Matrix

The promoter peaks matrix exploits annotation-based information about promoter peaks to identify relevant genes in the GAM. The identification of promoter peaks in GAGAM v1.2 has changed compared to GAGAM v1.0 [10].

The golden rule applied in GAGAM is that *a gene in a cell is active if, and only if, its promoter peak is accessible.* This rule reduces the set of interesting genes to consider when constructing a GAM.

Let us denote with $P^{p}\subseteq P$ the subset of peaks in the dataset D annotated as promoters (i.e., $\mathrm{PL}\left(p\right)=\mathrm{prom}\;\forall p\in P^{p}$) and with $D_{|P^{p}\left|\times |C\right|}^{p}$ the submatrix of D including only rows associated to promoter peaks.

GAGAM v1.2 constructs a binary matrix ${\mathrm{GP}}_{|G^{p}\left|\times \right|P^{p}|}$ associating the set of genes with active promoter peaks ($G^{p}$) to their related peaks. To associate a promoter peak to a gene, GAGAM v1.2 considers overlapping an enlarged gene body region, including 500 bp before the TSS with the peak region. This value was selected as an estimation of the mean peak length, but other values can be used depending on the available data.

The identification of promoter peaks has changed compared to GAGAM v1.0 [10]. GAGAM v1.2 employs two rules to define the promoter peaks. It relies on the promoter signature of the cCREs tracks, but it also considers the peaks overlapping the genes’ TSS.

The motivation for this choice is twofold.

First, even if the cCREs track information is curated and well-organized, it is still a prediction, and therefore, it has intrinsic limitations linked to partial prediction accuracy. Indeed, some genes in the NCBI RefSeq Genes annotation appear to have no promoter signature near the TSS (Figure 3). So, to not lose possible information about the activity of this category of genes, it is convenient to label the peaks overlapping their TSS as prom.

Second, the genes are directional (meaning they organize along an intended direction for the transcriptional process to occur, where, for example, the promoter regions precede the coding region). Still, the cCREstracks signatures are not. This can bring problems when assigning the promoter peaks to the right genes. For example, in genomic regions where two divergent genes overlap, a prom peak could be wrongly assigned to the most overlapping gene (Figure 4). Therefore, GAGAM v1.2 employs the innate TSS-gene association to correctly set the prom peaks to the genes.

Based on this, the promoter peaks matrix is a binary matrix computed as:(3)$$P_{|G^{p}\left|\times |C\right|}={\mathrm{GP}}_{|G^{p}\left|\times \right|P^{p}|}\times D_{|P^{p}\left|\times |C\right|}^{p}$$

This matrix is a GAM based on accessibility data for the subset of genes in the dataset exhibiting promoter peaks. In this way, GAGAM leverages available knowledge on transcriptional regulatory regions to select, among the entire group of genes considered in the scATAC-seq dataset, only the active ones, based on the modeling assumption that promoter accessibility is a necessary condition for gene regulation and transcription.

### 3.3. Promoter-Enhancer Co-Accessibility Matrix

The co-accessibility matrix accounts for the connections between promoters and enhancers. It is the second most important contribution and employs the enhD peaks.

GAGAM v1.2 leverages Cicero [7] to calculate the co-accessibility of the peaks. The co-accessibility represents how couples of peaks are simultaneously accessible in the cells, expressing it in a range between 0 and 1. This calculation *“connects regulatory elements to their putative target genes”* [7], meaning it may find connections between the promoters and the distal regulatory regions where different elements like Transcriptional Factors (TF) bind and enable transcription. The first step to compute this matrix is to calculate the co-accessibility from the scATAC-seq data with the Cicero function run_cicero (for the explanation of the calculation, refer to [7]). The result is a list of peaks couples with their co-accessibility value (ca) and distance (*d*) in the form $\mathrm{conn}=(p^{1},p^{2},ca,d)$. However, the co-accessibility alone does not carry out the information on the gene regulatory regions. Connections found by the algorithm do not describe gene regulation links, so many are unnecessary for this model.

GAGAM v1.2 selects only couples of promoter-enhancer peaks, i.e., couples with $p^{1}\in P^{p}$ and $p^{2}\in P^{e}$ (or vice versa), with $P^{e}\subseteq P$ representing the subset of peaks in the dataset D annotated as enhancers (i.e., $\mathrm{PL}\left(p\right)=\mathrm{enhD}\;\forall p\in P^{e}$). Compared to the GAGAM v1.0 [10], GAGAM v1.2 is more precise in selecting couples of peaks, making sure to select all the combinations without repetition.

GAGAM v1.2 considers only strong and relevant peaks connections, which are more likely to represent actual regulatory links. This is done thanks to two criteria: the distance between connected peaks and the strength of the connections, represented by the two parameters $d_{th}$ and $c_{am}$. These parameters derive from the Cicero pipeline co-accessibility calculation function [7], a well-established co-accessibility analysis pipeline.

Co-accessibility scores tend to have low values. GAGAM v1.2 uses the $ca_{m}$ parameter to determine the strongest connections (couples with $ca\geq ca_{m}$) and filter out connections that are not relevant to the gene to reduce noise. $c_{am}$ is the mean value of all the non-zero co-accessibility scores. This threshold is automatically computed based on co-accessibility data from each specific dataset, adapting the selection of the considered connections. This provides a pretty strict selection (10% at most) of connections and prevents inconsistencies linked to inter-dataset differences since the co-accessibility results can vary a lot between experiments.

The distance $d_{th}$ is a threshold to define the span of the genomic region where connections are most probably actual direct regulation links. It allows selecting only connections within a relevant distance range in the genome (i.e., $d\leq d_{th}$). Unfortunately, the literature lacks a clear indication for choosing the value of this parameter. Therefore, GAGAM v1.2 employs the exact value the Cicero guidelines suggest (i.e., 30,000 bp) [7].

At this point, the analysis identifies co-accessible couples of genomic regions, selects couples composed of a promoter and an enhancer, and among them, keeps only promoter-enhancer couples lying within the $d\leq d_{th}$ range of 30,000bp and whose accessibility score was above the mean value of all the co-accessibility scores above zero ($ca_{m}$). This is the starting point to build the co-accessibility contribution for GAGAM computation.

To calculate the co-accessibility matrix C, GAGAM uses three matrices. First, the binary matrix ${\mathrm{GP}}_{|G^{p}\left|\times \right|P^{p}|}$ previously defined in Section 3.2 and associating genes with promoter peaks. Second, the matrix ${\mathrm{PE}}_{|P^{p}\left|\times \right|P^{e}|}$ associating promoter peaks and enhancer peaks. The elements of this matrix are the co-accessibility values ca of couples of peaks available in the list produced by Cicero and 0 otherwise. Third, the matrix $D_{|P^{e}\left|\times |C\right|}^{e}$ is a submatrix of D including only rows associated with enhancer peaks.

Based on this, the co-accessibility matrix is as follows:(4)$$C_{|G^{p}\left|\times |C\right|}={\mathrm{GP}}_{|G^{p}\left|\times \right|P^{p}|}\times {\mathrm{PE}}_{|P^{p}\left|\times \right|P^{e}|}\times D_{|P^{e}\left|\times |C\right|}^{e}$$

### 3.4. Exon Peaks Matrix

The third contribution to the final GAGAM relies on exon accessibility within the gene coding region.

GAGAM v1.2 considers the contribution of the peaks inside the gene body more strictly than GAGAM v.1.0 [10]. As aforementioned, intronic regions of the gene bodies can contain regulatory regions. These do not necessarily relate to the gene containing them. Thus, instead of labeling all the peaks inside the gene body as intra, GAGAM v1.2 distinguishes between exons and introns. Therefore, as discussed in Section 3.1, only the peaks overlapping exons take the exon label and constitute the third contribution to the GAS.

As detailed in Section 5, the current GAGAM computation uses a single promoter for each coding region. For this reason, the calculation of accessible exons’ contribution to gene activity is referred to as the selected promoter.

Let us denote with $P^{i}\subseteq P$ the subset of peaks in the dataset D annotated as intragenic (i.e., $\mathrm{PL}\left(p\right)=\mathrm{intra}\;\forall p\in P^{i}$) and with $D_{|P^{i}\left|\times |C\right|}^{i}$ the submatirx of D including only rows associated to exon peaks. GAGAM v1.2 constructs a matrix ${\mathrm{GI}}_{|G^{p}\left|\times \right|P^{i}|}$ associating genes with active promoter peaks ($G^{p}$) to their related exon peaks. This matrix only considers genes with active promoter peaks to follow the GAGAM golden rule (Section 3.2). The reason is that some of the identified exon peaks could be part of genes that do not have a promoter peak. Moreover, it could happen that given a gene region inside a cell, exon peaks could be accessible even if the promoter peak is not.

The number and the length of exons vary a lot between the genes of a genome. Thus, statistically, there will be more peaks overlapping exons of a long gene, meaning it might have a higher score after its length. To prevent this bias, GAGAM employs a strategy from the GeneScoring [27] method to compute the elements of GI. It weighs the contribution of the exon peaks with an exponentially decaying function of their distance from the TSS (i.e., ${\mathrm{GI}}_{g,p}=a\cdot e^{-\frac{d}{5000}}$ where a=1 if p⊆g, 0 otherwise and *d* is the distance of the peak from TSS). In this way, very long genes are not over-represented because the most crucial part of the gene’s activity is near the promoter. Therefore, the peaks near it are weighted more. GAGAM v1.2 improves the peak distance calculation by considering the distance of the center of the peak to the TSS, not just the nearest base.

Based on this, the exon peaks matrix is a matrix computed as:(5)$$I_{|G^{p}\left|\times |C\right|}={\mathrm{GI}}_{|G^{p}\left|\times \right|P^{i}|}\times D_{|P^{i}\left|\times |C\right|}^{i}$$

### 3.5. Final Matrix Construction

After the computation of the three contributions above, the final step is constructing the GAGAM v1.2. As done in [10], this work defines two combinations of contributions for GAGAM construction, one with all three contributions (i.e., $w_{p}=1$, $w_{c}=1$, and $w_{e}=1$), and one without the exon peaks part (i.e., $w_{p}=1$, $w_{c}=1$, and $w_{e}=0$)). This allows us to assess the exon peaks matrix contribution and, thus, the role of exon accessibility to the analysis performance.

Moreover, GAGAM v1.2 adds another element to the final construction. GAGAM v1.0 [10] lacked normalizing the final matrix scores. Since there is not a reliable normalization method suited for these contributions, GAGAM v1.2 employs the normalization approach proposed by Cicero (for the explanation of the method, refer to [7] and its documentation). This ensures at least a normalization between the different impacts of the three contributions. To establish the actual effects of the normalization, Section 4 shows the results on the normalized GAGAM v1.2.

## 4. Results and Discussion

This section explains the strategy proposed in this work to validate GAGAM v1.2, the metrics chosen to measure performance, the analysis pipelines, and the datasets employed. Finally, it shows and discusses the results obtained.

### 4.1. Validation Strategy

GAGAM models gene activity based on scATAC-seq data, providing an enhanced tool to study cellular heterogeneity based on epigenomic data and to build a robust comparison with transcriptomics. Based on this consideration, following the same approach proposed in [10], the proposed validation strategy aims to assess the capabilities of GAGAM v1.2, considering performance under two lenses: (1) the capability of correctly identifying cellular heterogeneity and (2) the enhancement of biological significance for multi-omics studies.

The first validation step refers to (1), which evaluates GAGAM v1.2 capabilities to identify cellular heterogeneity. This is measured by quantifying the consistency of the results obtained with GAGAM-based clustering analysis versus those obtained with established approaches on the same datasets. Classification performance measures and comparisons rely on Adjust Rand Index (ARI) [29] and Adjust Mutual Information (AMI) [30], whose usage is thoroughly discussed in Section 4.2.

The second validation step refers to (2). It is to prove that GAGAM v1.2, and, in particular, its model-based selection of genes to consider for the analysis, have biological meaning, intended in two aspects: the intrinsic informativeness of the GAGAM and its capability to support a comparison with transcriptomics. The first aspect is measured by quantifying the capability of the analysis to detect significant differences in gene activity among identified clusters. This measurement relies on the Residual Average Gini Index (RAGI) metric (see Section 4.2). The second aspect is investigated with a preliminary comparative analysis between Differentially Active (DA) and Differentially Expressed (DE) genes (see Section 4).

### 4.2. Metrics Definition

The validation strategy proposed in Section 4.1 relies on the unsupervised clustering of cells based on the genes selected for GAGAM construction. Clustering analysis outputs clusters of cells modeling the heterogeneous types of cells in a cell population. The first step includes evaluating whether GAGAM-based clustering analysis yields clusters that correctly represent cell heterogeneity. Comparing results from clustering classification based on GAGAM v1.2 with ground truth is necessary. Intending to do so, there are two scenarios: (i) the dataset includes cell labels from pre-existing classifications. Thus, each cell has a label identifying its cell type, or (ii) the dataset does not include cell labels, so there is no ground truth to compare. In the first scenario, it is possible to directly compare GAGAM-based clustering classification with the cell-type labels leveraging ARI and AMI metrics to assess similarity. Since there is no reference classification in the second scenario, ARI and AMI cannot be used as in the previous case. An alternative method is to employ clustering-based labels obtained from the raw scATAC-data analysis as ground truth and then use ARI and AMI to measure the results’ similarity. Otherwise, it is possible to use the RAGI metric to assess and compare the intrinsic capability of different classification methods to distinguish cells based on activity (or accessibility) profiles.

#### 4.2.1. ARI and AMI

This work relies on the ARI [29] and AMI [30] metrics from information theory to measure similarity in classification results. These two metrics are often employed for this type of evaluation. In particular, ref [31] uses them for benchmarking, and they are the metrics employed to evaluate GAGAM v1.0 [10], which supports the direct comparison of the new results with those previously produced. ARI and AMI similarity measures range between 0 and 1, where 1 is a perfect match, and 0 is complete dissimilarity. The main difference between the two is that ARI performs better when the ground truth clustering has large clusters with equal cell numerosity. In contrast, AMI performs better when the ground truth clustering yields unbalanced clusters in size, including small clusters [41]. Considering this, even though AMI is preferable for most cases considered in this work, the results rely on both metrics.

#### 4.2.2. RAGI

The RAGI [31] is based on the hypothesis that good clustering results should exhibit the high activity of marker genes in specific clusters and activity of housekeeping genes in all the cells from every cluster. A good GAM should convey meaningful biological information that should translate into a difference in the activity patterns across clusters between the two sets of genes. Computing the difference between the Gini index of markers and housekeeping genes respectively, the RAGI can measure the intrinsic capability of GAGAM-based clustering analysis to distinguish groups of cells with the same functional profile (i.e., the activity of specific marker genes), relying on internal control (i.e., activity the housekeeping genes). On the other hand, RAGI is highly dependent on the genes chosen to calculate it. Indeed, the RAGI algorithm requires two lists of genes, housekeeping genes, and marker genes, which are not trivial to select because there is no universal definition of housekeeping genes or marker genes. Therefore, the selection of housekeeping genes is partly arbitrary and could bring errors to the calculation. This is even worse in the case of marker genes, which are highly debatable [42] and should be decided after the cell types present in the dataset, which is only partially defined. Therefore, it is essential to carefully choose the right set of marker genes strictly related to the dataset sample. In this work, the list of housekeeping genes is based on the literature, particularly on [43] for human cells and on [44] for murine cells. On the other hand, the marker genes for the human Peripheral Blood Mononuclear Cell (PBMC) datasets are based on a curated list of markers for PBMC classification from [45], while the marker genes for the mouse brain dataset were defined based on [46,47].

### 4.3. Datasets

This work analyzes GAGAM v1.2 performance on six datasets (see Table 1 and Table 2). Three of these datasets supported the validation of GAGAM v1.0 [10]; the others are new. Five out of six datasets are from the 10XGenomic platform [24] since it provides high-quality datasets generated by various technologies. The datasets shared with [10] consist of a collection of respectively 5335 (*10X v1.0.1 PBMC* [48]) and 4623 (*10X V2.0.0 PBMC* [49]) cells from human Peripheral Blood Mononuclear Cells (PBMC) samples. Additionally, this study also employed the dataset of bone marrow (*Buenrostro2018*) from [50], consisting of 2034 cells with provided cell-type classification. The three new datasets consist of one scATAC-seq dataset of mouse brain with 7729 cells (*10X V2.1.0 Brain* [51]), and two from the single cell Multiome technology, consisting respectively of 10,691 cells (*10X Multiome Controller PBMC* [52]) and 10,970 cells (*10X Multiome Chromium X PBMC* [52]) from human PBMC samples. These two datasets provide scATAC-seq and scRNA-seq data, supporting the preliminary comparison between the activity and expression of genes. None of the three new datasets have cell labels. Therefore, their ARI and AMI evaluations leverage raw scATAC-seq clustering-based labels.

### 4.4. Pipelines

For clustering analysis and preliminary comparison with transcriptomics, the pipeline of choice is Monocle3 [53], given its simplicity, the straightforward workflow, and the fact that Cicero GAM (see Section 2) is dependent on it (Figure 1). The standard Monocle workflow starts with a GAM, performs Principal Component Analysis (PCA), visualizes the cells in 2D using UMAP [54], and, most importantly, conducts cell clustering. However, Latent Semantic Indexing (LSI) is an alternative approach to the usual PCA, especially for scATAC-seq data. Thus, since GAGAM v1.2 is a direct manipulation of scATAC-seq data, it makes sense to employ LSI.

Concerning performance evaluation, ARI and AMI computation relies on the R package ARICODE [55]. In contrast, RAGI computation is based on a custom Python script (see **Data Availability Statement**).

### 4.5. Results

This Section provides the results of the validation strategy proposed in Section 4.1.

Results refer to the following comparisons:GAGAM v1.2 *versus* GAGAM v1.0 [10], over the old datasets;GAGAM v1.2 *versus* Cicero [7] and GeneScoring [27], over the new datasets.

Besides considering both GAGAM v1.0 and GAGAM v1.2, the experimental setup considers two versions of GAGAM v1.2:GAGAM1 v1.2 created using the complete GAGAM workflow (i.e., $w_{p}=1$, $w_{i}=1$, and $w_{c}=1$)GAGAM2 v1.2 constructed considering only the promoter peaks and the co-accessibility (i.e., $w_{p}=1$, $w_{i}=0$, and $w_{c}=1$).

#### 4.5.1. Peak Labeling

As a first insight into the GAGAM v1.2 construction process, Table 3 shows the final number of labeled peaks per dataset, focusing on their labeling and the final number of genes employed for GAGAM construction.

#### 4.5.2. Clustering Similarity

The first step of the validation strategy described in Section 4.1 aims to evaluate GAGAM v1.2 capability of correctly identifying cellular heterogeneity. At this aim, this section compares the similarity of results obtained with GAGAM-based clustering analysis *versus* results obtained with other approaches. Measures and comparisons rely on the ARI and AMI metrics (see Section 4.2).

Figure 5 and Figure 6 report AMI and ARI scores for all methods under analysis over the considered datasets. Comparisons rely on ground truth labels. For example, clustering labels are compared with pre-existing cell type labels for the *Buenrostro2018* dataset [50]. In contrast, clustering labels are compared with the raw scATAC-seq clustering results for the other datasets. Figures display the ARI and AMI scores for the two GAGAM v1.2 versions (GAGAM1 v1.2 and GAGAM2 v1.2, respectively, patterned bars) against GAGAM v1.0 from [10] (lighter blue tones) for the three Old datasets (left), and Cicero and GeneScoring (darker blue tones) for the three New datasets (right). Results show that both versions of GAGAM v1.2 perform better than GAGAM v1.0, Cicero, and GeneScoring. However, GAGAM v1.2 offers higher performances than GAGAM v1.0, with some significant improvements. In particular for *10X V2.0.0 PBMC* where the AMI of GAGAM1 v1.2 has a gain of about 23%. Only on *10X V2.1.0 Brain*, GeneScoring has a higher metrics value than GAGAM. However, as discussed in [10], even if GeneScoring has, sometimes, higher performances of AMI and ARI, it has abysmal results of the RAGI, lowering the impact of these higher metrics. Generally, both versions of GAGAM v1.2 score higher in ARI and AMI metrics than clustering results from the scATAC-seq data. This suggests that GAGAM v1.2 can identify cellular heterogeneity at least at the same level as the direct scATAC analysis.

#### 4.5.3. Biological Significance

The second step of the validation strategy described in Section 4.1 aims to assess GAGAM v1.2 biological significance, proving the model-based selection of genes to consider and their elaboration enhances informativeness and biological meaning. This is measured by quantifying the capability of the analysis to detect significant differences in gene activity among identified clusters. This measurement relies on the RAGI metric (see Section 4.2).

Figure 7 and Figure 8 show the results of RAGI computation, assessing the information content of the GAMs (GAGAM1 v1.2, GAGAM2 v1.2, GAGAM1 v1.0, GAGAM2 v1.0, Cicero and GeneScoring GAMs, respectively) compared to other clustering classifications. Following the same logic from ARI and AMI results presentation (see Section 4.5.2), the two versions of GAGAM v1.2 are shown with patterned bars, the two versions of GAGAM v1.0 from [10] in lighter blue tones for the three Old datasets (left), and Cicero and GeneScoring are shown in darker blue tones for the three New datasets (right). To perform RAGI computation (as described in Section 4.2.2), this work employed three different sets of curated marker genes, one for each type of tissue in the datasets (human PBMC, human bone marrow, mouse brain) and two different sets of housekeeping genes, one for each species in the datasets (human and mouse, respectively).

Results for biological significance organize into two groups. The first group (Figure 7) leverages RAGI scores to compare informativeness between the clustering results obtained with the different GAMs. The second group (Figure 8) leverages RAGI scores to compare the clustering results of each GAM with classification labels from pre-existing cell type labels (in the case of the *Buenrostro2018* dataset [50]) or from clustering-based labels obtained from clustering analysis on raw scATAC-seq data. This way, all methods are evaluated against the same partition to understand which GAM is the most biologically consistent.

In general, results show how GAGAM v1.2 has good RAGI scores, which are either better or comparable to scores of other methods. Improvements here are less significant than those on ARI and AMI scores. Only the *10X v1.0.1 PBMC* dataset shows lower performance than the scATAC-seq clustering results. Consistently with what is shown in [10], although GeneScoring’s clustering results are consistent with the ground truth classification (as indicated by ARI and AMI), RAGI results highlight how GeneScoring GAM performs poorly as for both RAGI scores (Figure 7 and Figure 8) and the respective *p*-values over the tolerable threshold of 0.05 (data not shown, see **Data Availability Statement**), meaning they are not statistically significant. This highlights the importance of evaluating the GAMs performances on both aspects of clustering similarity to identify cellular heterogeneity and biological significance.

Overall, the results show that GAGAM1 v1.2 (the version with all three contributions) generally performs better, or at least equally, than GAGAM2 v1.2, highlighting that considering all three contributions is relevant to maximize performance. This emerges more clearly in this work than in [10].

#### 4.5.4. Preliminary Comparison with Transcriptomics

Finally, the proposed validation strategy aims to support the comparison between epigenomic and transcriptomic data for multi-omics studies. As discussed, multi-omics datasets allow for comparing the GAGAM results with the scRNA-seq data. This work proposes a preliminary qualitative comparison between the activity and expression of DA genes identified by GAGAM v1.2 and scRNA-seq clustering analysis. At first, this process performs differential analysis on the GAGAM v1.2 clustering. This yields the most DA genes for each cluster. Then, to investigate if the activity and the expression of specific genes are consistent, it provides two types of visualizations: violin plots and a heatmap, showing the activity and the expression levels of the most DA genes. It is possible to investigate if the activity and the expression of specific genes are consistent.

As an example, Figure 9 and Figure 10 shows the results on both 10X multiome datasets, *10X Multiome Controller PBMC* and *10X Multiome Chromium X PBMC*, considering the GAGAM1 v1.2 (the version including all three contributions). This plot visualizes, for each cluster, the expression level distribution of the top DA considered genes across cells. Even if not the same, the activity and expression profiles tend to be consistent. Anyway, these preliminary results are promising. For example, Figure 10 shows how the activity profile of the gene MS4A1 (a well-known marker for PBMC cells) is highly consistent with its expression profile, meaning it is both active and expressed in the same cells.

Figure 11 and Figure 12 provides heatmaps, again for both 10X multiome datasets, that allow a similar comparison, extending it to a broader panel of DA genes (specifically the top 10 DA genes per cluster). In addition, the heatmaps show the activity (on the top) and the expression (on the bottom) by cluster throughout all the datasets, allowing a more comprehensive investigation and comparison of differential activity and expression profiles, respectively. Again, Figure 11 shows overall consistency between the activity and expression of these genes throughout the cells of the dataset.

However, as expected, there is no complete concordance, especially in the case of the *10X Multiome Chromium X PBMC* (which shows more discrepancies), but this is a preliminary qualitative investigation. Further investigation of these differences between the activity and expression of genes could be informative and will follow. In fact, of great interest will be to understand what the relationships are between highly correlated and poorly correlated activity-expression genes. One explanation of the discrepancies between activity and expression can derive from the different time scales between the two biological processes, meaning that the variation in gene expression has a faster dynamic than the epigenetic changes. Moreover, the scATAC-seq has a much lower resolution than the scRNA-seq, making it difficult to completely grasp the actual epigenetic state. Even if this is a preliminary and qualitative analysis, it demonstrates how the content of GAGAM v1.2 has a biological meaning that supports comparison with the gene expression. This is not trivial and establishes the GAGAM as a powerful tool to assess the relationship between accessibility and expression. More generally, a tool for multi-omics data analysis. Here the paper shows this first qualitative analysis that allows investigation of the general relationship of activity and expression patterns of some genes. However, as discussed in Section 3.1, the modular nature of GAGAM will allow easy follow-up analyses. Specifically, it will be interesting to inspect if different accessibility patterns of each contribution affect the expression levels, discriminating between highly expressed transcripts vs. moderate or low expressed transcripts. In addition, an interesting point would be to calculate the ratio between the promoter and exon accessibility to investigate its possible correlation with the gene expression subsequently. This will surely be a future improvement, sensibly extending GAGAM data exploration and interpretation capability.

## 5. Conclusions

The presented work focuses on the capabilities of GAGAM to yield a biologically significant and understandable way to compute gene activity and compare it with gene expression levels. In conclusion, GAGAM v1.2 improves on the previous version of GAGAM v1.0, maintaining its novelty as a method to obtain a GAM from scATAC-seq data. Furthermore, it is based on a model-driven approach leveraging genomic annotations of genes and functional elements.

GAGAM v1.2 improves the correct and thoughtful labeling of the peaks. Specifically, it enhances the precision of the *prom* labeling, the proper association with the right genes (covering some limit cases), the focus on exons instead of on the entire gene body, and the appropriate identification of *enhD* peaks from intronic regions.

Experimental results demonstrate how GAGAM v1.2 improved performances compared to the previous version, confirming its dominance towards Cicero and GeneScoring. Furthermore, GAGAM v1.2 improved its scores for all the metrics employed, showing that the GAGAM v1.2-based clustering is comparable with the one obtained from the raw scATAC-seq data, supporting the claim that GAGAM v1.2 can identify cellular heterogeneity.

In addition, GAGAM is a suitable method to interpret accessibility data in general. Indeed, since it employs genes as features, it allows analyzing scATAC-seq data through well-studied and investigated concepts like marker genes. The same analysis would not be possible with raw accessibility data. The analysis of the RAGI score supports this claim and highlights the gene activity differences between marker and housekeeping genes. The direct comparison with transcriptomics further and more coherently supports this. Indeed, as a further step, the activity of the genes is compared with their expression on the same cells, highlighting consistency between the two biological levels.

The GAGAM approach stems from a radical innovation of genomic annotation of accessibility data toward multimodal single-cell data analysis. The GAGAM genomic model captures non-trivial contributions of different genomic regions to the potential functional regulation of transcriptions at a gene, interpreting and weighing them in a separate and specific way, and rationally organizing them in the final GAS computation. This fully interprets the final GAS and supports biologically meaningful accessibility data analysis and direct comparison with gene expression levels.

On the other hand, the non-triviality of the comparisons between activity and expression data highlights the importance of interpretability and biological significance of GAS computation. The genomic model at the core of GAGAM computation works and will further develop in this direction, to characterize the complex relationship between genomic accessibility and transcription. The modularity of final matrix computation allows the separate handle of the contribution of well-defined parts, quantifying the impact of each contribution to the final GAGAM computation and subsequent analysis. For example, it will be exciting to investigate whether different gene expression levels correspond to specific patterns in overall gene activity or any specific individual contribution of GAGAM. Lastly, the modular nature of GAGAM will enable the seamless extension of the genomic model in a modular way, progressively including new contributions in the final GAGAM computation and the possibility to select which ones to include every time. For instance, this could ease the implementation of a GAGAM version, which will include different isoforms of genes. Generally, it provides a tool for the bioinformatician to tailor the GAGAM application for specific purposes, explicitly evaluating and transparently choosing which contributions to consider.

Besides the improvement and modular expansion of the genomic model, another direction for improvement for GAGAM is toward usability. In the direction of existing analysis pipelines, accessibility and usability are essential to allow the bioinformatics community to leverage the GAGAM approach’s capabilities easily. Thus it would be valuable to develop the GAGAM pipeline to include an improved data exploration and interpretation capability. Furthermore, future works aim to allow easier customization of the analysis parameters to allow users to adapt the GAGAM pipeline to their specific analyses. In this direction, this work already provides the complete analysis pipeline and guidance to reproduce the proposed experiments (see *Data Availability statement*).

In conclusion, GAGAM v1.2 is a promising and reliable way to interpret scATAC-seq data, which focuses on the accessibility of the genes and their regulatory elements, with the potential to act as a direct link between epigenomic and transcriptomic, proving to be a promising tool, especially for multi-omic data analysis.

## Figures and Tables

**Figure 1 genes-14-00115-f001:**
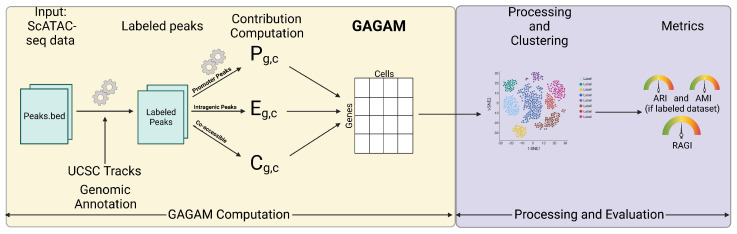
**Workflow for computation and evaluation of GAGAM.** The workflow starts with scATAC-seq data, and labels the peaks with the help of genomic annotations and UCSC tracks. Then it computes the three contributions forming GAGAM. GAGAM is evaluated and compared to other GAMs through clustering experiments with three well-established metrics: ARI [29], and AMI [30] (if the dataset is labeled) or RAGI [31] (if the dataset is not labeled).

**Figure 2 genes-14-00115-f002:**
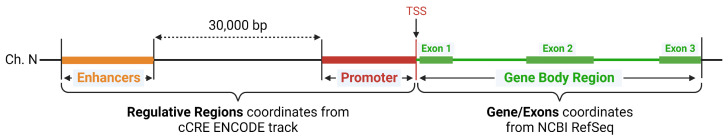
**Genomic Model.** The genomic model consists of the coordinates of all the genomic regions related to the gene. The gene body region (green) comes from the NCBI RefSeq Genes annotations. The regulative regions, i.e., Promoter (in red) and Enhancers (in orange) come from the cCREs ENCODE tracks.

**Figure 3 genes-14-00115-f003:**
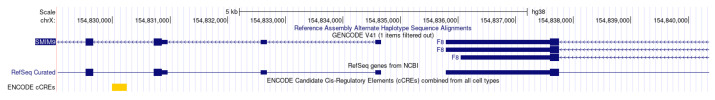
**Genomic region of SMIM9 gene.** The figure shows there are no prom signature regions, in the proximity of the TSS of SMIM9, in the cCREs track, meaning it would be left out from the gene activity computation if looking only at Promoter signatures.

**Figure 4 genes-14-00115-f004:**
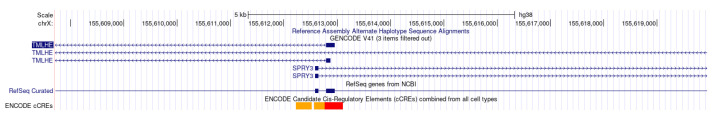
**Genomic region of TMLHE and SPRY3 genes.** In this case, the promoter signature (red) is an ambiguous position between the two divergent genes, and due to some overlapping, the prom peaks could be assigned to the wrong genes.

**Figure 5 genes-14-00115-f005:**
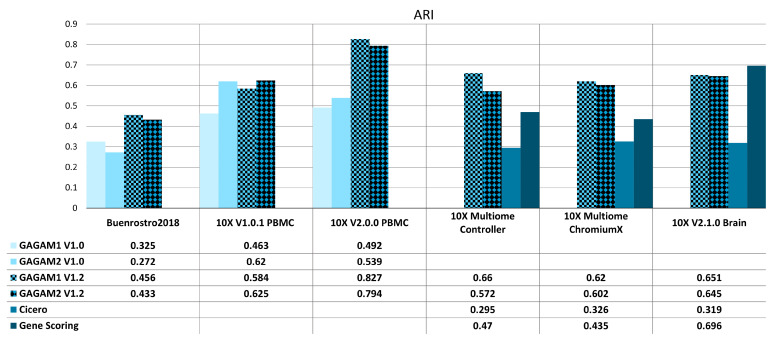
ARI results for both Old (**left**) and New (**right**) datasets. The Old datasets are compared with GAGAM v1.0 from [10], while the New datasets are with Cicero and GeneScoring.

**Figure 6 genes-14-00115-f006:**
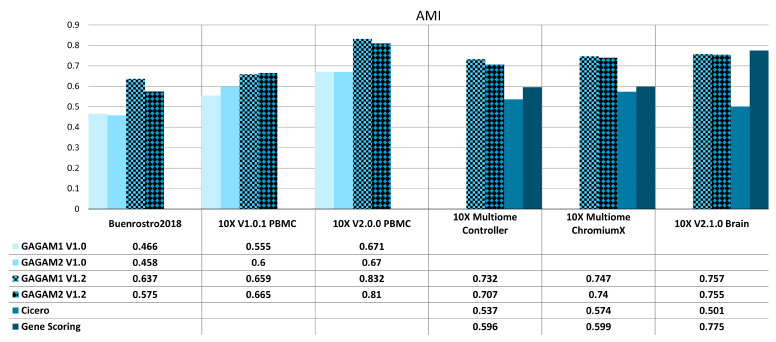
AMI results for both Old (**left**) and New (**right**) datasets. The Old datasets are compared with GAGAM v1.0 from [10], while the New datasets are with Cicero and GeneScoring.

**Figure 7 genes-14-00115-f007:**
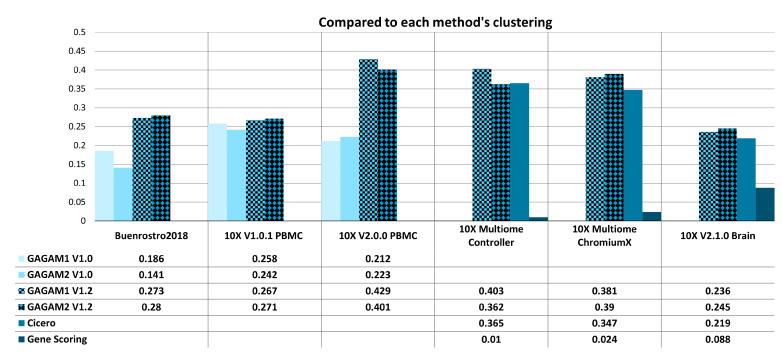
RAGI results of both Old (**left**) and New (**right**) datasets when compared to each method’s clustering. The Old datasets are compared with the previous GAGAM v1.0 from [10], while the New datasets are with Cicero and GeneScoring.

**Figure 8 genes-14-00115-f008:**
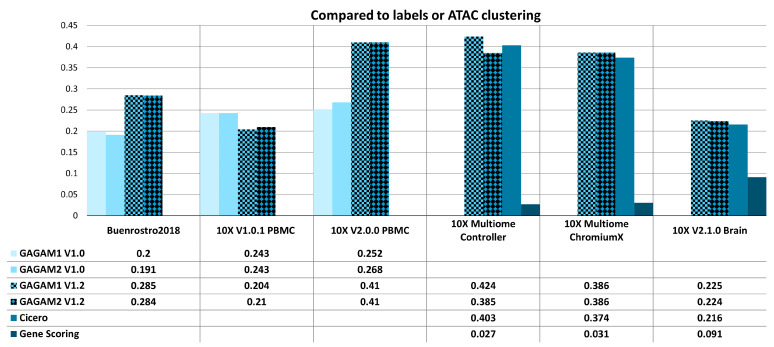
RAGI results of both Old (**left**) and New (**right**) datasets when compared to labels or ATAC clustering. The Old datasets are compared with the previous GAGAM v1.0 from [10], while the New datasets are with Cicero and GeneScoring.

**Figure 9 genes-14-00115-f009:**
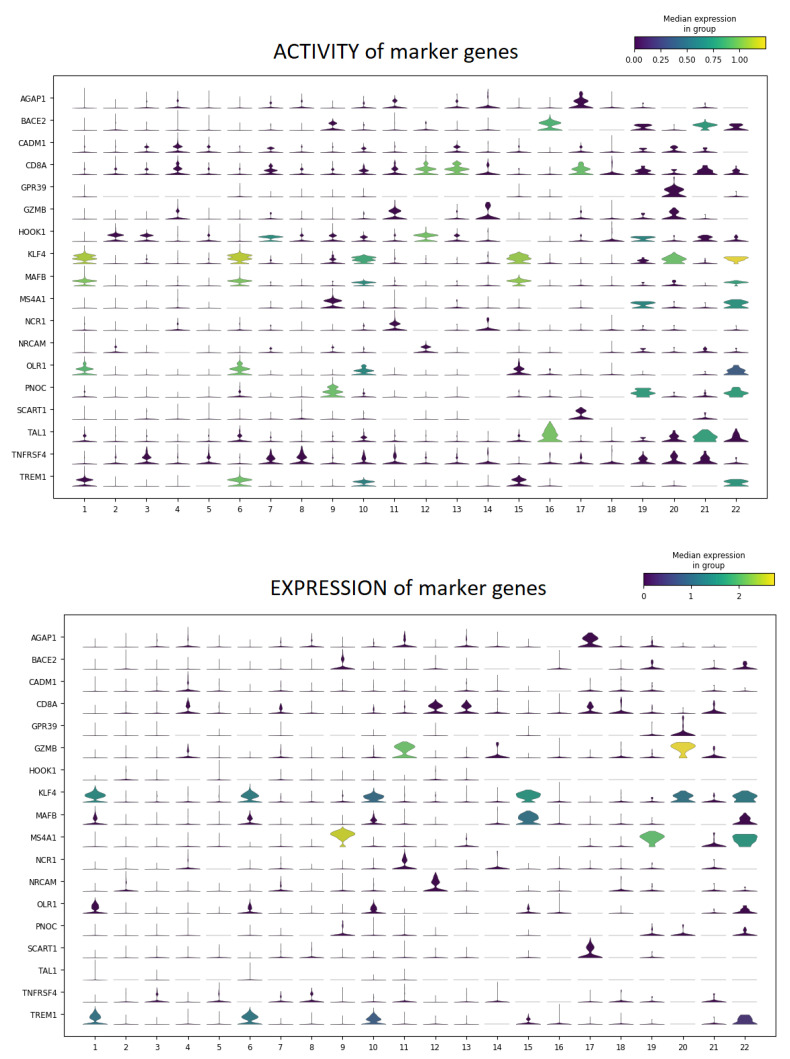
Violin Plots for the top DA genes for each cluster, of their Activity (on the **top**) and Expression (on the **bottom**), for the *10X Multiome Controller PBMC* dataset. The two figures show a general coherence (especially for genes like MS4A1, AGAP1, and KLF4) between the activity and expression profiles on the same cells, showing consistency between the two biological levels.

**Figure 10 genes-14-00115-f010:**
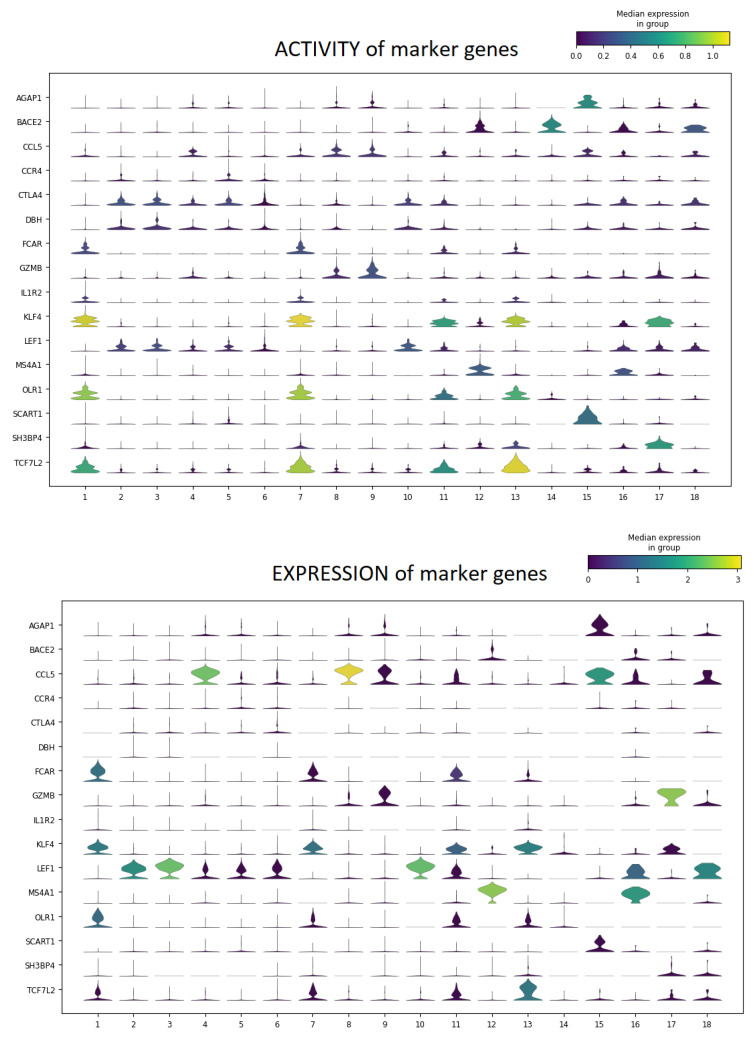
Violin Plots for the top DA genes for each cluster, of their Activity (on the **top**) and Expression (on the **bottom**), for the *10X Multiome Chromium X PBMC* dataset. The two figures show a general coherence (especially for genes like MS4A1, AGAP1, and KLF4) between the activity and expression profiles on the same cells, showing consistency between the two biological levels.

**Figure 11 genes-14-00115-f011:**
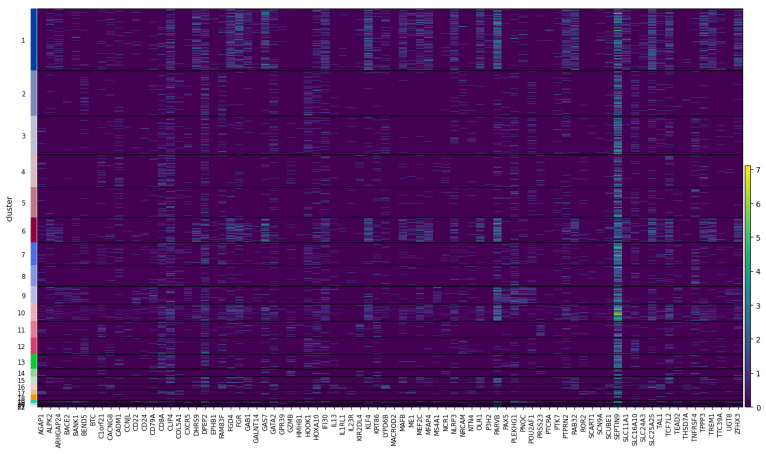
Heatmap for the top 10 DA genes, showing their Activity (on the **top**) and Expression (on the **bottom**) profiles throughout the whole *10X Multiome Controller PBMC* dataset. Again, one can notice repeated patterns between the two.

**Figure 12 genes-14-00115-f012:**
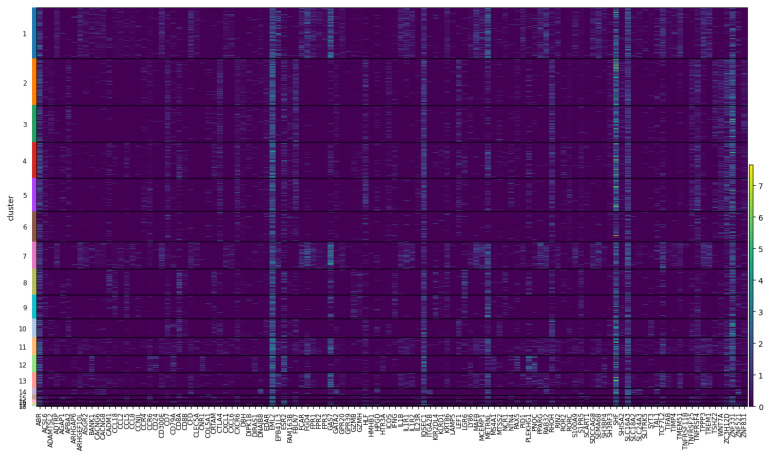
Heatmap for the top 10 DA genes, showing their Activity (on the **top**) and Expression (on the **bottom**) profiles throughout the whole *10X Multiome Chromium X PBMC* dataset. Again, one can notice repeated patterns between the two, even if, in this dataset, the discrepancies are more and the correlation is less clear.

**Table 1 genes-14-00115-t001:** Old datasets employed.

Dataset	Species	Tissue	Cells Labels	Reference
*10X v1.0.1 PBMC*	Human	PBMC	No	[48]
*10X V2.0.0 PBMC*	Human	PBMC	No	[49]
*Buenrostro2018*	Human	Bone Marrow	Yes	[50]

**Table 2 genes-14-00115-t002:** New datasets employed.

Dataset	Species	Tissue	Cells Labels	Reference
*10X Multiome Controller PBMC*	Human	PBMC	No	[52]
*10X Multiome Chromium X PBMC*	Human	PBMC	No	[52]
*10X V2.1.0 Brain*	Mouse	Brain Cortex	No	[51]

**Table 3 genes-14-00115-t003:** Numbers of peak labels.

Dataset	Total Peaks	*Prom* Peaks	*enhD* Peaks	*Exon* Peaks	Final Genes
*10X v1.0.1 PBMC*	97,124	11,651	1940	7763	11,488
*10X V2.0.0 PBMC*	135,373	15,720	79,481	3650	13,298
*Buenrostro2018*	491,232	22,847	190,298	127,667	14,905
*10X Multiome Controller PBMC*	115,179	15,528	67,790	3207	13,230
*10X Multiome Chromium X PBMC*	111,743	15,438	66,211	3064	13,153
*10X V2.1.0 Brain*	176,651	14,662	50,784	9768	13,023

## Data Availability

The data presented in this study are openly available in NCBI GENE Expression Omnibus (GEO) with accession number GSE96769, and from the freely available 10XGenomic platform [24] datasets. All the code and material created are available at https://github.com/smilies-polito/GAGAM Accessed on 29 December 2022.

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
