# Peer review of "GAGAM v1.2: An Improvement on Peak Labeling and Genomic Annotated Gene Activity Matrix Construction"

_genes, 2022, doi:10.3390/genes14010115_

Round 1

Reviewer 1 Report

Authors designed a updated version GAGAM v1.2 to interpret scATAC-seq data by associating regulatory elements with genes, which could be anticipated as a promising tool for linking epigenomics and transcriptomics. They proved its power from diversity of metrics such as ARI, AMI and RAGI. However following questions need to be solved.

1.      Although their results suggested the method show superior performance over several benchmark methods, their contribution should be more clearly illustrated. According to my opinion, their most significant contribution is an updated GAGAM version with a re-defined GAM matrix. What different and effective strategies were proposed in this study compared with the routine pipeline for scATAC-seq annotation?

2.      They said ‘GAGAM v1.2 considers overlapping an enlarged gene body region, including 500bp before TSS’, ‘keeps only couples with ca > cam, and d≤dth’. How did they determine these parameters? How did these parameters affect the final analysis performance? Is it possible to give a more flexible choice in their tool for a good user-friendliness?

3.      They defined the activity score as: GAGAM = wp · P + wc · C + we · E. how the weighted determined? The influence caused by the training dataset should be estimated before practicing their tool.

4.      Instead of the description on regulatory element background, the authors were suggested give more detail about the problems on the prom,enhance and gene association analysis.   

Reviewer 2 Report

In the current manuscript, the authors developed GAGAM V1.2, a better version of GAGAM v1.0. It combines accessibility with genome annotations, thereby providing a better understanding of different accessible regions on the overall gene activity.

Major comments:

1.       Alternative transcripts show cell type specific expression during development and differentiation. Some of the important cell type specific genes exhibit alternative transcripts expression using alternative promoters. For some genes, alternative promoters are located downstream of TSS. Can GAGAM v1.2 identify accessibility at alternative promoters when they are present intragenic?

In section 2.4, the gene scoring method uses peaks at exons only and not introns. Peaks near promoters are weighted more. In case active alternative promoter usage, the peaks at alternative promoters will not be scored. What could be the possible explanation for this?

1.       In the case of exon peaks matrix, GAGAM v1.2 relies on the exon accessibility and considers the peaks inside the gene body. However, the downregulated or low expressed genes show some levels of accessibility in the exonic regions. The accessibility increases in case upregulated genes. What could be the threshold used to distinguish these two scenarios?

2.       Related to Figure 9. Please provide the violine plot representation of highly cluster specific gene accessibility in one violine plot instead of separate violine plots. And provide the violine plot representation of expression levels of the same gene set using scRNA-Seq. This will provide clearer information along with individual violine plot representation. And it will be useful to understand to what extent scATAC-Seq analysis by GAGAM v1.2 could be used for clustering and the extent of similarity with gene activity in the clusters.

3.       Related to Figure 10, Many genes show accessibility but not expression across the clusters. Some of them are RAB32, BECE2, BEND5, NCR1, SPRY2, TAL1. Overall, some genes show good correlation with their expression levels but not all. Please provide details on the discrepancy observed.

4.       Please provide the comparison analyses for more than one sample in the case of Figure 9 and Figure 10 using GAGAM v1.2 and also GAGAM v1.0.  

5.       It would be more prominent to see the correlation of the top 10 highly specific genes of each cluster between GAGAM v1.2 analysis of scATAC-Seq and compare it with scRNA-Seq. Please provide the analysis for more than one sample.

6.       Does the ratio of accessibility at promoters and exons have any implications on the levels of gene expression? Can this be addressed using GAGAM v1.2? If so, what could be the accessibility patterns at promoters, coding exons and enhancers for highly expressed transcripts vs moderate or low expressed transcripts?   

Round 2

Reviewer 1 Report

All my questions have been well answered!